# The Bioengineered Combo Dual-Therapy CD34 Antibody-Covered Sirolimus-Eluting Coronary Stent in Patients with Chronic Total Occlusion Evaluated by Clinical Outcome and Optical Coherence Tomography Imaging Analysis

**DOI:** 10.3390/jcm10010080

**Published:** 2020-12-28

**Authors:** Recha Blessing, Majid Ahoopai, Martin Geyer, Moritz Brandt, Andreas M. Zeiher, Thomas Münzel, Philip Wenzel, Tommaso Gori, Zisis Dimitriadis

**Affiliations:** 1Department of Cardiology, Cardiology I, University Medical Center of the Johannes Gutenberg University, 55131 Mainz, Germany; recha.blessing@unimedizin-mainz.de (R.B.); majid.ahoopai@unimedizin-mainz.de (M.A.); martin.geyer@unimedizin-mainz.de (M.G.); moritz.brandt@unimedizin-mainz.de (M.B.); tmuenzel@uni-mainz.de (T.M.); wenzelp@uni-mainz.de (P.W.); tommaso.gori@unimedizin-mainz.de (T.G.); 2Center for Thrombosis and Hemostasis (CTH), Johannes Gutenberg University, 55131 Mainz, Germany; 3Department of Cardiology, Center of Internal Medicine, Goethe University Frankfurt, 60590 Frankfurt, Germany; zeiher@em.uni-frankfurt.de; 4German Center for Cardiovascular Research (DZHK), Partner Site Rhine-Main, 55131 Mainz, Germany

**Keywords:** Combo^®^ DTS, chronic total occlusion, optical coherence tomography

## Abstract

We sought to determine the effects of the use of a Bioengineered Combo Dual-Therapy CD34 Antibody-Covered Sirolimus-Eluting Coronary Stent (Combo^®^ DTS) in patients with chronic total occlusion (CTO) by evaluating clinical outcomes and by performing an optical coherence tomography (OCT) analysis. We retrospectively analyzed data from 39 patients who had successfully undergone OCT-guided revascularization of a CTO being treated with a Combo^®^ DTS. Clinical assessment, angiography (with quantitative coronary angiography analysis) and OCT examination were performed at baseline and at follow-up. The median follow-up period was 189 days, ranging from 157 to 615 days. At follow-up, revascularization was required due to angiographic restenosis in 40% (14 of 35) of patients. OCT analysis detected neointima proliferation in 23 (76.6%) patients. Neointima formation was often associated with microvessels in 18 patients (60%). Neoatheroslcerosis was observed in 2 (6.6%) patients. Malapposition was found in 4 patients (13.3%), and stent fractures were found in 11 patients (36.6%). Rate of strut coverage was 96.3% at follow-up. In conclusion, the implantation of a Combo^®^ DTS after successful CTO recanalization was associated with a restenosis rate of 40% despite good stent implantation at baseline, proven by OCT. Neointima formation was found as a main contributor to restenosis. Nevertheless, we observed a low rate of major cardiovascular events in our follow-up.

## 1. Introduction

About 15–20% of patients with coronary heart disease suffer from a chronic total occlusion (CTO) of a coronary artery. CTO is defined as a lesion with a 100% stenosis and a thrombolysis in myocardial infarction (TIMI) grade 0 that exists for more than three months [1]. The therapy and treatment success of CTO lesions has improved significantly in recent years due to new devices and catheter techniques [1,2,3]. Despite these advances, CTO lesions have a high risk of in-stent stenosis after successful recanalization. The mechanisms of restenosis following recanalization are multifactorial and often remain unclear. Risk factors and predictors that have already been described are clinical, angiographic and procedural characteristics as well as the chosen procedural techniques (e.g., dissection re-entry technique with subintimal wire tracking) [4,5,6,7,8]. Likewise, stent type—and here in particular bare metal stents (BMSs) and first-generation drug eluting stents (DESs)—is an important determinant of a future incidence of restenosis [1] High rates of stent strut malapposition and incomplete stent strut coverage after CTO percutaneous coronary intervention (PCI) are observed, which is related to stent thrombosis [9,10,11] The Bioengineered Combo Dual-Therapy CD34 Antibody-Covered Sirolimus-Eluting Coronary Stent (Combo^®^ DTS; ORBUSNEICH MEDICAL., Ft. Lauderdale, FL, USA) is designed to enable faster endothelial healing by combining traditional components of DESs with an anti-CD34 antibody coating. The abluminal layer consists of a biodegradable polymer coated with sirolimus in order to prevent excessive neointima formation. In addition, the luminal layer consisting of anti-CD34 antibodies is intended to promote rapid endothelial healing [12,13,14]. 

The Combo^®^ DTS showed encouraging results after percutaneous coronary intervention (PCI) in different patient groups with high rates of stent strut coverage [15,16,17,18]. As delayed endothelialization after CTO PCI is known, we investigated the use of the Combo® DTS, which is said to have a rapid endothelialization and high rate of stent coverage. However, the impact of Combo® DTS use has not been studied in patients after successful revascularization of CTO lesions. In this study we investigated by retrospective data analysis the efficacy of the Combo® DTS in patients with CTO lesions based on clinical outcome and quantitative coronary angiography analysis (QCA). Stent implantation and vascular response after Combo^®^ DTS implantation was controlled by using optical coherence tomography (OCT) imaging.

## 2. Methods

### 2.1. Study Object

The Combo^®^ DTS is made of stainless steel with a dual-helix design. The stent struts have a width of 90 μm and a thickness of 100 μm. The abluminal bioabsorbable polymer is layered with sirolimus (5 μm/mm); the luminal side is covered with an anti-CD34 antibody layer. According to the manufacturer’s specifications, sirolimus is released over 30 days, and the polymer is absorbed over 90 days.

The antiproliferative effect of the sirolimus layer leads to a reduction of the restenosis rate. It also causes delayed reendothelialization, which could increase the risk of stent thrombosis in the long term. The abluminal anti-CD34 antibody layer binds to endothelial progenitor cells (EPCs) in order to improve endothelial healing [14,15,16,17,18,19,20,21,22,23].

### 2.2. Study Design and Study Population

From September 2018 to March 2019 at the University Medical Center Mainz, we retrospectively analyzed data from 39 patients who had successfully undergone OCT-guided revascularization of a CTO due to stable angina with noninvasive ischemia testing or due to acute coronary syndrome. All patients were treated with a Combo^®^ DTS.

Clinical assessment, angiography (with quantitative coronary angiography analysis (QCA)) and OCT examination were performed at baseline and at follow-up surveillance angiography. The clinical follow-up was performed 6 months after stent implantation as part of routine clinical care. Patients with a restenosis in the first follow-up were re-examined after 6 months with additional coronary angiography. In consideration of ischemic and bleeding risks, the patients were treated with dual antiplatelet therapy for 12 months after complex PCI. One patient on anticoagulants received aspirin at the time of index PCI followed by 12 months therapy with clopidogrel [1].

### 2.3. Clinical Endpoints and Definitions

Successful CTO recanalization was defined as restoration of TIMI flow grade 3 and <30% stenosis within the treated CTO segment by visual estimation and quantitative coronary angiography analysis (QCA). The primary endpoint was target vessel failure (TVF, defined as the presence of total re-occlusion with TIMI flow grade 0, restenosis (>50%)) and target vessel revascularization (TVR, defined as any revascularization within the treated vessel). Restenosis of the CTO vessel was defined as stenosis >50% by visual estimation and late lumen loss >50% quantified by QCA (Philips Healthcare, Andover, MA, USA). Major adverse clinical events were all-cause death and myocardial infarction.

Myocardial infarction was defined according to the Fourth Universal Definition of Myocardial Infarction Guidelines [24].

### 2.4. Optical Coherence Tomography Image Analysis

OCT imaging was performed directly after PCI and in the follow-up to evaluate stent implantation and vascular healing. OCT was carried out with the OCT imaging system ILUMIEN™ OPTIS™ (St Jude Medical, MN) and with a Dragonfly™ OPTIS™ imaging catheter (Abbott Medical, Westford Massachusetts, US) using the nonocclusive technique. The qualitative and quantitative analysis of OCT imaging was performed by two observers with the QCU-CMS Software (research edition, Medis Medical Imaging Systems; Leiden, the Netherlands). Quantitative measurements were analyzed at 1 mm intervals. 

OCT analysis included minimal, maximal and average stent and lumen diameter and area. 

Stent expansion and apposition were assessed by using stent asymmetry (minimum lumen diameter/maximum lumen diameter > 0.7), stent eccentricity ((maximum stent diameter minus minimum stent diameter)/maximum stent diameter > 0.7) and no significant residual stenosis causing minimal lumen area <4 mm^2^. Incomplete strut apposition (ISA) was defined as stent area minus lumen area [25,26,27,28]. 

Neointima was defined as the tissue between the vessel lumen border and the luminal border of the stent struts, and it was classified as homogenous or heterogenous. Homogenous neointima was defined as tissue with uniform optical patterns. Heterogenous neointima was defined as tissue with different patterns [29,30]. Neointima thickness was calculated as stent area minus in-stent lumen area [23,31,32].

Microvessels were defined as signal-poor vesicular structures <200 μm which could be followed over several consecutive frames [31]. Evagination was defined as a cavity between luminal vessel contour and apposed struts [33]. Neoatherosclerosis was defined as atherosclerotic changes (presence of lipid or calcified plaque) within the neointima [23]. Stent fracture was defined as strut fracture with stent deformation and intraluminal protruding stent strut [28,34]. The evaluation of strut coverage was done in all frames, and all visible struts were analyzed.

### 2.5. Statistical Analysis

Continuous normally distributed data are presented as mean and standard deviation and compared using the Student’s *t*-test; values not normally distributed are presented as median and minimum and maximum values and compared using the Mann–Whitney U-test. Categorial data are presented as numbers and frequencies and compared using the chi square test. A two-sided *p*-value of <0.05 was considered to be significant. The statistical analyses were performed using SPSS (Version 23, IBM SPSS Statistics, SPSS Inc., Chicago, IL, USA).

## 3. Results

Thirty-nine patients were analyzed in the study and treated with Combo^®^ DTS after successful revascularization of a CTO. Follow-up data of 35 patients were available; 4 patients refused surveillance coronary angiography and were lost to follow-up. The median follow-up period was 189 days, ranging from 157 to 615 days. During this period, coronary angiography was performed to investigate the result after CTO recanalization. The majority of patients underwent surveillance coronary angiography after 6 months, as recommended at our center after complex PCI. Because the study was retrospective, the follow-up was carried out later in a few patients. 

### 3.1. Clinical and Angiographic Parameters

Clinical and angiographic baseline characteristics are listed in Table 1. We investigated a complex study population with a high J-CTO Score (2.49 ± 0.5) and a high rate of multivessel coronary vessel disease (CVD). Thirty-eight CTO lesions were de-novo lesions; one CTO lesion was an in-stent-stenosis. More than two-thirds of the CTO lesions were located in the right coronary artery. A retrograde approach was performed in 23.1% of the cases. Stent implantation was controlled by OCT imaging. At baseline, there was no malapposition or stent fracture.

### 3.2. Quantitative Coronary Angiography Analysis

At follow-up, revascularization was required in 40% (14 of 35) of patients with angiographic restenosis (confirmed by QCA). Quantitative coronary angiography analysis is presented in Table 2.

One patient (2.9%) had a total re-occlusion with TIMI flow grade 0. The remaining patients had restenosis of <50% in the treated vessel. Eleven patients (31.4%) with in-stent restenosis were treated with a drug eluting balloon (DEB). Three patients (8.6%) were treated with implantation of a drug-eluting stent (DES). In the next control, the patients who were treated with DESs showed a good angiographic result without need for further treatment. In 4 of the 11 patients who were treated with DEBs, the DEB had to be repeated. The overall major adverse clinical event rate was 8.5% (3/35). One patient died of cancer. One patient with myocardial infarction in the follow-up was treated with coronary bypass surgery due to rapid progression of coronary heart disease. One patient presented with myocardial infarction and acute heart failure due to left main disease after 615 days. Both patients showed good angiographic results of the treated vessel after revascularization. None of the clinical and procedural characteristics were associated with the incidence of restenosis at six months.

### 3.3. Results of Optical Coherence Tomography 

The results of the OCT imaging at baseline and in the follow-up are shown in Table 3. Stent asymmetry was 0.73 ± 0.17, and stent eccentricity was 0.7 ± 0.09. Minimal lumen area was 2.11 mm^2^ ± 1.23 at follow-up.

Neointima proliferation was detected in 23/30 patients; 14/23 showed homogenous and 9/23 showed heterogenous neointima. Neointima formation was often associated with microvessels (18/30). Neoatheroslcerosis was observed in 2/30 patients. Malapposition was found in 4/30 patients, and stent fractures were found in 11/30 patients. A stent thrombosis was detected in only one patient with a stent fracture. Rate of strut coverage was 96.3% at follow-up. Results are presented in Table 3 and Table 4.

## 4. Discussion

A CTO lesion was found in 15–20% of patients with chronic stable coronary artery disease. The improvement in quality of life, clinical symptoms and prognostic benefits after successful recanalization of a CTO has been demonstrated in several studies [1,35,36,37]. One of the main challenges after CTO PCI is the occurrence of restenosis. The rate of restenosis can be reduced by the use of newer-generation DESs compared to bare metal stents, but it remains as high as 10–15% [1,35,36]. Various risk factors for restenosis after successful recanalization have been described. Important clinical characteristics include sex, age and diabetes mellitus. Angiographic and procedural characteristics for restenosis are longer lesion length, higher number of implanted stents (>3) and smaller stent diameter. Recent studies focused on catheter techniques and showed that the dissection re-entry technique with subintimal wire tracking retrograde access is also associated with an increased restenosis rate compared to intraluminal wire tracking. Subintimal wire tracking probably occurs much more frequently than expected, but can only be reliably detected by intracoronary imaging [4,5,6,7].

In a previously published study by Geyer at al., possible predictors for target vessel failure after recanalization of chronic total occlusions were investigated in a collective of 93 patients at our center. The incidence of target vessel failure was 15.1% after successful chronic total occlusion intervention. Female gender and reduced TIMI flow immediately after recanalization were found to be predictors for TVF. Target vessel failure was a combined endpoint, defined as re-occlusion (in 7.5% of the patients), restenosis (defined as lumen loss >50% in the CTO vessel quantified by QCA) (in 11.8% of the patients) and target vessel revascularization (in 5.4% of the patients). This rate of TVF is lower than the rate we found (15.1% vs. 40%). It should be noted that the complexity of the study collectives differs. The J-CTO score was higher and different catheter techniques were used in our collective (e.g., retrograde approach and subintimal wire tracking) [8]. 

Stent diameter and length as well as the number of implanted stents also increased restenosis rate after CTO recanalization [5,6,7]. In 41% (16/39) of the patients in our collective, three or more DESs were implanted; the mean lesion length was 39.08 ± 15.65 mm. The J-CTO score uses a lesion length greater than 20 mm and calcification within the CTO segment to calculate the difficulty category. Considering these risk factors for TVR, the patients in our collective fulfill several predictors for TVF. In our collective, stent eccentricity <0.7 was the only OCT measurement associated with restenosis. Stent eccentricity <0.7 is known to be related to a higher rate of clinical events, malapposition, stent thrombosis and target vessel revascularization. These findings could be confirmed in our collective. One reason for our finding could be an increased calcification of CTO lesions [38,39,40]. 

The clinical and angiographic outcome of the Combo^®^ DTS has already been investigated in many studies, some of them involving large patient populations and long follow-up periods. Implantation of a Combo^®^ DTS seems to be safe and feasible. The rate of clinical events is low. 

Of the patients treated with a Combo^®^ DTS, target lesion revascularization occurred in 1.4–7.5%, and stent thrombosis occurred in 0.5–0.8% [21,41,42]. There are few studies published which compare the efficacy of Combo^®^ DTSs to other newer-generation DESs. Haude et al. showed comparable rates of clinical events and target vessel revascularization between Combo^®^ DTSs to TAXUS™ Liberté™ DESs in a five-year follow-up [22]. It should be noted that the TAXUS™ Liberté™ DES is a first-generation DES. A meta-analysis already showed a better clinical and angiographic outcome in patients with CTO PCI after implantation of second-generation DESs compared to first generation DESs [43].

Based on this promising data after implantation of a Dual-Therapy Endothelial Progenitor Cell Capturing Sirolimus-Eluting Stent, the present study investigated, for the first time, clinical and angiographic outcomes after implantation of Combo^®^ DTS in CTO lesions in a complex collective. Compared to the mentioned rates of target lesion revascularization after standard PCI, we found a substantially higher rate in our collective after CTO PCI (1.4–7.5% vs. 40%).

So far there is only limited data from studies using intravascular imaging to investigate CTO PCI. Based on OCT imaging, delayed DES coverage, neointima formation and high rates of malapposed stent struts were described after CTO PCI. Studies with intravascular imaging suggest that optimal stent implantation in a CTO lesion is challenging [10,11,33]. For this reason, we used OCT imaging after baseline PCI to verify the result of a good stent implantation and expansion and at follow-up to examine possible reasons for restenosis.

Neointima hyperplasia was found in 76.6% (23/30) of our patients, and was the main cause for restenosis in our collective. Other studies have also found enhanced neointima formation after Combo^®^ DTS implantation in standard PCI visualized by OCT imaging. Lee et al. showed an increase in neointima formation in the first nine months, and Saito et al. found a greater neointimal hyperplasia thickness in the Combo^®^ DTS group compared to the DES group (Xience, Abbott Vascular, Santa Clara, CA, USA) after one year (0.18 mm vs. 0.107 mm, *p* < 0.001) [23,44].

A recently published meta-analysis of five studies was conducted to analyze whether the DTS was superior to standard DESs. One year, TLR was higher in a DTS arm compared to a second-generation DES arm [45]. The outcome after stent implantation depends mainly on the stent design, polymer coating and drug type. Thicker stent struts especially seem to promote neointima formation as well as procoagulant and proinflammatory elements. Compared to other second-generation drug-eluting stents, Combo^®^ DTSs have thicker stent struts (82 μm vs. 100 μm) [46]. Anti-CD34 antibody labeling is intended to attract EPCs in order to improve endothelial healing. EPCs are multipotent adult stem cells that have the potential to differentiate in both endothelial cells and smooth muscle cells. They are intended to provide an endogenous repair mechanism due to the promotion of rapid endothelialization. On the other hand, higher numbers of circulating EPCs correlate with increased numbers of restenosis [47]. The role of EPCs after CTO PCI is unclear. Our data suggest augmented EPC recruitment due to anti-CD34 antibody labeling in the DTS promoted neointima formation. Possible pathomechanisms could be an excessive differentiation of EPCs in the proinflammatory environment of a CTO lesion as well as enhanced attraction of myeloid cells to the lesion side [48].

Larger prospective and randomized studies are needed to confirm our results of a high restenosis rate and increased neointima formation in a complex patient collective after CTO PCI and Combo^®^ DTS implantation.

## 5. Conclusions

Our study is the first to investigate clinical and angiographic outcome after Combo^®^ DTS implantation in CTO lesions. The implantation of a Combo^®^ DTS after successful CTO recanalization was associated with a restenosis rate as high as 40% despite good stent implantation at baseline proven by OCT. Neointima formation was found as a main cause of restenosis. Nevertheless, we observed a low rate of major cardiovascular events in our follow-up.

## Figures and Tables

**Table 1 jcm-10-00080-t001:** Demographic and procedural characteristics.

Clinical Characteristics	All Patients (*n* = 39)	Restenosis Present (*n* = 14)	No-Restenosis Present (*n* = 21)	*p*-Value
Demographics Characteristics				
Age, yrs	67.21 ± 11.57	64.29 ± 13.29	68.81 ± 10.78	1
Male	34 (87.2)	12 (30.7)	18 (46.2)	0.29
Diabetes mellitus	10 (25.6)	5 (12.8)	3 (7.7)	0.22
Hypertension	38 (97.4)	14 (35.9)	20 (51.3)	1
Hyperlipidemia	31 (79.5)	9 (23.7)	18 (46.2)	0.22
Current smoking	12 (30.8)	4 (10.2)	7 (17.9)	1
Multivessel CVD	37 (94.8)	13 (33.3)	20 (51.3)	0.48
Glomerular filtration rate	75 (20–99)	76 (23–99)	77 (20–95)	0.96
LVEF	55 (30–61)	55 (32–61)	55 (34–61)	0.89
BMI	29.31 ± 3.94	28.64 ± 2.4	30.67 ± 4.26	0.11
Procedural characteristics				
RCA	29(74.4)	11 (28.2)	16 (41.0)	1
LAD	5 (12.8)	2 (5.12)	2 (5.1)	1
LCX	5 (12.8)	1 (2.6)	3 (7.7)	0.63
J-CTO Score	2.49 ± 0.5	2.64 ± 0.49	2.33 ± 0.48	0.07
Antegrade approach	30 (76.9)	10 (25.6)	17 (43.6)	0.68
Retrograde approach	9 (23.1)	4 (10.3)	4 (10.3)	0.68
Number of stents ≤ 3	16 (41)	8 (20.5)	6 (15.4)	0.09
Number of stents	2 (1–4)	3 (1–4)	2 (1–4)	0.14
Stent diameter (mm)	2.75 (2.5–4)	2.75 (2.5–3.5)	2.8 (2.5–4)	0.19
Stent length (mm)	24 (13–34)	24 (18–33)	23 (13–34)	0.5
Clopidogrel	29 (74.3)	8 (57.1)	17 (80.9)	0.2
Prasugrel	6 (15.3)	4 (28.5)	2 (9.5)	0.1
Ticagrelor	3 (7.6)	1 (7.2)	2 (9.5)	0.8
Oral anticoagulation/Clopidogrel	1 (2.5)	1 (2.5)	0	0.4

Values are *n* (%), median (interquartile range) or mean ± SD; yrs = years, CVD = cardiovascular disease, LVEF = left ventricular ejection fraction, BMI = body mass index, RCA = right coronary artery, LAD = left anterior descending coronary artery, LCX = left circumflex coronary artery.

**Table 2 jcm-10-00080-t002:** Quantitative coronary angiography analysis.

QCA Measurements	Baseline PCI (*n* = 39)	Follow-Up PCI (*n* = 35)
Minimum lumen diameter (mm)	2.03 ± 0.43	1.54 ± 0.63
Maximum lumen diameter (mm)	3.64 ± 0.71	3.6 ± 0.81
Mean lumen diameter (mm)	2.81 ± 0.53	2.25 ± 0.46
Stenosis (%)	17.7 (3–25)	34 (13–100)
Late lumen loss (mm)		0.66 ± 0.47

Values are *n* (%), median (interquartile range) or mean ± SD; PCI = percutaneous coronary intervention.

**Table 3 jcm-10-00080-t003:** Results of quantitative OCT assessment at follow-up.

OCT Measurements	All Patients (*n* = 30)	Restenosis Present (*n* = 13)	No Restenosis Present (*n* = 17)	*p*-Value
**Baseline**				
Number of struts	419.97 ± 164.32	385.38± 147.57	445.94 ± 179.5	0.31
CTO length (mm)	39.08 ± 15.65	36.58 ± 13.48	42.24 ± 17.04	0.23
Stent volume (mm^3^)	282.75 ± 158.83	238.83 ± 142.92	315.56 ± 170.14	0.19
Stent-lumen volume (mm^3^)	5.77 ± 16.38	2.77 ± 9.69	8.02 ± 20.27	0.35
Residual stenosis (%)	31.56 ± 13.55	25.38 ± 12.93	36.11 ± 12.87	0.03
Maximum stent asymmetry	0.73 ± 0.17	0.75 ± 0.08	0.72 ± 0.22	0.65
Maximum stent eccentricity	0.7 ± 0.09	0.69 ± 0.12	0.66 ± 0.09	0.58
Stent asymmetry > 0.3	17 (43.6)	4 (28.6)	12 (66.7)	0.04
Stent eccentricity < 0.7	15 (38.5)	6 (42.9)	9 (50)	0.8
ISA (mm)	0.19 ± 0.22	0.13 ± 0.19	0.24 ± 0.24	0.17
**Follow-up**				
Number of struts	394.19 ± 165.8	385 ± 154.61	397.2 ± 178.5	0.84
Minimum lumen area (mm^2^)	2.11 ± 1.23	1.5 ± 0.89	2.53 ± 1.29	0.57
Stent volume (mm^3^)	263.83 ± 167.4	247.02 ± 170.73	275.7 ± 169.23	0.19
Stent-lumen volume (mm^3^)	93.18 ± 158.04	91.8 ± 117.72	94.15 ± 184.87	0.30
Maximum stent asymmetry	0.41 ± 0.15	0.41 ± 0.13	0.41 ± 0.17	0.96
Maximum stent eccentricity	0.89 ± 1.19	0.66 ± 0.07	1.06 ± 1.6	0.37
Stent asymmetry > 0.3	24 (61.5)	12 (85.7)	13 (61.9)	0.37
Stent eccentricity < 0.7	27 (71.1)	12 (85.7)	11 (52.4)	0.02
Neointima thickness (mm)	1.18 ± 0.59	1.33 ± 0.42	1.08 ± 0.69	0.25
uncovered struts (%)	3.72 ± 5.19	2.62 ± 3.89	4.49 ± 5.93	0.31

Incomplete stent apposition (ISA), values are *n* (%), median (interquartile range), or mean ± SD.

**Table 4 jcm-10-00080-t004:** Qualitative OCT assessment at follow-up.

OCT Assessments	All Patients (*n* = 30)	Restenosis Present (*n* = 13)	No Restenosis Present (*n* = 17)	*p*-Value
Evagination	8 (26.6)	4 (13.3)	4 (13.3)	0.76
Stent fracture	11 (36.6)	7 (23.3)	4 (13.3)	0.13
Malapposition	4 (13.3)	2 (6.6)	2 (6.6)	0.87
Neointima	23 (76.6)	12 (40)	11 (36.6)	0.02
Homogenous	14 (46.6)	5 (16.6)	9 (30)	0.7
Heterogenous	9 (30)	7 (23.3)	2 (6.6)	0.01
Microvessels	18 (60)	9 (30)	9 (30)	0.37
Neoatherosclerosis	2 (6.6)	1 (3.3)	1 (3.3)	0.9
Stent thrombosis	1 (3.3)	1 (3.3)	0	0.4

Values are *n* (%).

## Data Availability

The data presented in this study are available on request from the corresponding author. The data are not publicly available due to ongoing research.

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
