# Peer review of "The Bioengineered Combo Dual-Therapy CD34 Antibody-Covered Sirolimus-Eluting Coronary Stent in Patients with Chronic Total Occlusion Evaluated by Clinical Outcome and Optical Coherence Tomography Imaging Analysis"

_jcm, 2020, doi:10.3390/jcm10010080_

Round 1
Reviewer 1 Report
line 128
- As this is a study of a new device, all patients should be thoroughly followed up. Why was follow-up data available in only 35 patients? What happened to 4 patients?
line 129-: When was followup CAG performed? Was it variously performed during the long follow-up period?
More information is needed for the CTO lesion
- Were all de-novo lesions?
- What is the number of stent, mean diameter/length of stent that was implanted?
line 140 "At follow-up in 35.9% (14 of 39) patients with angiographic restenosis" Is the denominator 39 or 35?
line 160-: neointima proliferation was observed in 23 patients. Then the following analysis should be based on 23 patients.
Table 4: The full name of "homogen", "heterogen/" should be clarified
Overall, why did the authores compare the Combo DTS stent in CTO lesions? Is there a mechanistic explanation that the stent will be safe or efficious in CTO lesions?
It would be better if other "OCT based CTO PCI" studies were introduced, and the rate of stent fracture, malapposition etc. were compared.
Author Response
Dear Ms. Gu, Dear reviewer,
thank you for giving us the chance to revise our manuscript according to the insightful comments of the estimated reviewers. I hope you will see that we have carefully addressed all of the reviewer’s comments and I sincerely hope that based on the additional experiments that we added to the revised manuscript, our manuscript will be accepted for publication in Journal of Clinical Medicine.
According to the comments of reviewer 1 now additional information on the medical treatment of the patients with antiplatelet and anticoagulation therapy was added. A complete revision of the statistical analyzation with multilevel models was not feasible in the limited time of revision. In our manuscript the main point is the rates of restenosis with the COMBO DTS in CTO patients, additional OCT observations were used as possible explanations for our findings. In these cases, and after statistical advice we used t-test to. If mandatory we could rework our statistics, but we would need more time for this.
Reviewer 1:
I am not a statistician, but I think that the main limitation of this study is in the statistical analysis, in which the authors did not considered the cluster nature of the data, mandatory when oct data are analysed. I recommend to always use multilevel models when analysing multiple observations per patient, to make correct statistical inferences and not inflate statistical power. The statistical analysis should be repeated, in this paper, in which I am not an author (doi: 10.4244/EIJV9I1A23.) they could find examples.
We thank the reviewer for the evaluation of the manuscript and his helpful comment on the analyzation the data with multiple observations. We agree with him that multilevel analysis would improve the statistical value of the OCT analyzation data. The main aim of our study was to compare restenosis rate using the COMBO DTS in CTO lesions. Due to the limited number of patients we have carried out statistical processing in advance. A t-test or Mann-Whitney U-test was recommended to answer the main question of the study. Additional OCT analysis was performed to get further explanations for our findings with an increased rate of restenosis. Due to the very limited time of 5 days to revise the manuscript it was not possible to us to completely revise the statistical analyzation. If mandatory, we could repeat it with more time. We commented on this limitation in the discussion part of our manuscript
A table in which the dual antiplatelet therapy and the anticoagulant therapy is compared between the 2 group is mandatory.
The reviewer makes an important point. We reanalysed our data and added information of antiplatelet and anticoagulant therapy in Table 1. In our collective only one patient was treated with permanent anticoagulation with a good result in the long-term follow-up after 517 days.
Minor revisions:
The authors write in lane 80-83: "The clinical follow-up was performed 6 months after stent implantation as part of routine clinical care." Also the choice of dual antiplatelet therapies for 6 months is questionable. There are any data on new anticoagulants or Warfarin? I am not sure that this is suggested in any guidelines outside of clinical studies, The authors should justify both those affirmation with at least a few references.
We further clarified this comment in our manuscript. All of our patients were treated with dual antiplatelet therapy for 12 months after complex PCI, in consideration of their ischaemic and bleeding risks. One patient was on combined antiplatelet and anticoagulative treatment due to atrial fibrillation and anticoagulants received Rivaroxaban and 12 months therapy with clopidogrel following the PIONEER AF-PCI trail and current guidelines. We added this information in Table 1 as well as line 85-87.
Table 3: Stent Eccentricity <0.7 is the only univariate association with restenosis. This should be commented in discussion.
We took up this point in the discussion and commented on it in line 205-209.
line 171: references needed
We added the appropriate reference in line 171.
Reference 5 (Geyer M, Wild J, Hirschmann M, Dimitriadis Z, Münzel T, Gori T, Wenzel P. Predictors for target vessel failure after recanalization of chronic total occlusions in patients undergoing surveillance coronary angiography. Mainz: Johannes Gutenberg-Universität Mainz; 2020.) it is unclear where it was published.
The reference has been corrected.

Reviewer 2 Report
I read with great interest the study from R. Blessing et al. CTO are a very interesting challenge for interventional cardiology, and any new data on this field would be useful.
However, there are a few major issue that should be addressed before considering the paper.
Statistical analysis:
I am not a statistician, but I think that the main limitation of this study is in the statistical analysis, in which the authors did not considered the cluster nature of the data, mandatory when oct data are analyzed. I recommend to always use multilevel models when analyzing multiple observations per patient, to make correct statistical inferences and not inflate statistical power.
The statistical analysis should be repeated, in this paper, in which I am not an author (doi: 10.4244/EIJV9I1A23.) they could find examples.
A table in which the dual antiplatelet therapy and the anticoagulant therapy is compared between the 2 group is mandatory.
Minor revisions:
The authors write in lane 80-83 : "The clinical follow-up was performed 6 months after stent implantation as part of routine clinical care." Also the choice of dual antiplatelet therapies for 6 months is questionable. There are any data on new anticoagulants or Warfarin? I am not sure that this is suggested in any guidelines outside of clinical studies, The authors should justify both those affirmation with at least a few references.
table 3: Stent Eccentricity <0.7 is the only univariate association with restenosis. This should be commented in discussion.
line 171: references needed
reference 5 (Geyer M, Wild J, Hirschmann M, Dimitriadis Z, Münzel T, Gori T, Wenzel P. Predictors for target vessel failure after recanalization of chronic total occlusions in patients undergoing surveillance coronary angiography. Mainz: Johannes Gutenberg-Universität Mainz; 2020.) it is unclear where it was published.
Author Response
Dear Ms. Gu, Dear reviewer,
thank you for giving us the chance to revise our manuscript according to the insightful comments of the estimated reviewers. I hope you will see that we have carefully addressed all of the reviewer’s comments and I sincerely hope that based on the additional experiments that we added to the revised manuscript, our manuscript will be accepted for publication in Journal of Clinical Medicine.
According to the comments of reviewer 2 we added additional information on the CTO lesions in the manuscript and reworked our analyzations. We also added further studies on OCT and CTO lesions to the manuscript.
Reviewer 2:
line 128: As this is a study of a new device, all patients should be thoroughly followed up. Why was follow-up data available in only 35 patients? What happened to 4 patients?
We agree with the reviewer. All 35 patients were routinely scheduled for re-evaluation with a follow-up appointment at our cardiological clinic with an additional coronary angiography. Follow-up appointment was rejected by four patients without giving any further reasons. These four patients were classified as lost-to-follow-up.
line 129: When was follow-up CAG performed? Was it variously performed during the long follow-up period?
The follow-up examination including coronary angiography was performed following our clinical standard six month after COMBO-Stent implantation. The vast majority was examined after these 6 months, due to personal reasons of the patients or rescheduling of appointments some examinations were delayed.
More information is needed for the CTO lesion
- Were all de-novo lesions?
- What is the number of stents, mean diameter/length of stent that was implanted?
We added this information in line 142-143 and Table 1.
line 140 "At follow-up in 35.9% (14 of 39) patients with angiographic restenosis" Is the denominator 39 or 35?
The denominator is 35, we corrected the analysis and reworked the statistics.
line 160: neointima proliferation was observed in 23 patients. Then the following analysis should be based on 23 patients.
We changed our results and based the analyzation on the 23 patients.
Table 4: The full name of "homogen", "heterogen" should be clarified
We corrected Table 4 and added the corresponding definition in lines 113-116.
Overall, why did the authors compare the Combo DTS stent in CTO lesions? Is there a mechanistic explanation that the stent will be safe or efficious in CTO lesions?
CTO-lesions are complex stenosis with a very high rate of restenosis. In the current literature different types of stents are used in order to improve treatment of these lesions. In most of the studies a delayed stent strut coverage is described as one pathomechanism of restenosis. Here we wanted to use the advantage of the Combo DTS which is covered with an anti-CD34 antibody layer in addition to the established sirolimus layer. In other studies, with different collectives or coronary lesions the COMBO DTS was convincing with an enhanced rate of stent strut coverages and a safe use in PCI. Especially the ani-CD34 antibody layer is intended to enable rapid endothelialization which could make him favorable in CTO lesions. In order to prove or disprove this benefit, we analyzed the COMBO DTS.
It would be better if other "OCT based CTO PCI" studies were introduced, and the rate of stent fracture, malapposition etc. were compared.
After additional literature research and to the best of our knowledge there is only limited evidence on invasive reevaluation of CTO lesions, in particular, there are only a few studies that used OCT follow-up after successful CTO-PCI. As suggested, we added relevant studies to the manuscript (line 48-50). These studies predominantly describe a lower rate of stent strut coverages after CTO-PCI. We see in the standardized OCT follow-up examination one strength of our study to further understand the complexity of CTO lesions and to prevent the high restenosis rates. Here more studies are needed to decipher the pathomechanisms.

Round 2
Reviewer 1 Report
The answers fullfill my comments.
I have no further comments
Author Response
Thank you for the review. All questions from reviewer 1 were answered.
Reviewer 2 Report
I think the authors made a great work and the manuscript is improved. I, however, think that is mandatory to improve the statiscal analysis as i suggested and I agree with the authors that, as it is a very major revision, a few weeks are needed.
I am looking forward to revise the final manuscript.
Author Response
Thank you for the extra time you have given us so that we could consider and perform the proposed statistics. After consultation with the statistician of our department, who also contributed to the statistical processing of the data, it became clear that the data in this study could not be regarded as cluster data and that the proposed analysis was not possible. In order to confirm this statement, we asked an external statistical expert with many years of experience in the medical and cardiological sector for comment, Dr. Heinzel-Gutenbrunner. We attach her opinion. Dr. Heinzel also advises against such a statistical analysis with the study data.
We are really sorry for this inconvenience, but as not statistician we also though that this very specialized statistical analysis for cluster data, would be possible also with our data. We hope that you find our study with the current statistical analysis, which answers reliable the hypothesis, in what way the combo-stent is a better solution for CTO stenting, of a good quality and accept it for publication.
Please see the attachment
